# Efficacy and safety of Saireito (TJ-114) in patients with atrial fibrillation undergoing catheter ablation procedures: A randomized pilot study

Tetsuma Kawaji[1,2], Satoshi Shizuta[2]*, Hidenori Yaku[1], Kazuhisa Kaneda[1], Fumiya Yoneda[2], Shushi Nishiwaki[2], Munekazu Tanaka[1], Takanori Aizawa[2], Shun Hojo[1], Kenji Nakatsuma[1], Masashi Kato[1], Takafumi Yokomatsu[1], Shinji Miki[1], Koh Ono[2], Takeshi Kimura[3]

1 Department of Cardiology, Mitsubishi Kyoto Hospital, Kyoto, Japan, 2 Department of Cardiovascular Medicine, Graduate School of Medicine, Kyoto University, Kyoto, Japan, 3 Department of Cardiovascular Medicine, Hirakata Kohsai Hospital, Kyoto, Japan

* shizuta@kuhp.kyoto-u.ac.jp

**Data Availability Statement:** All relevant data are within the manuscript and its Supporting Information.

## Abstract

### Background

Early arrhythmia recurrences commonly occur after atrial fibrillation (AF) ablation because of irritability and inflammation of left atrium. We hypothesized that short-term use of Saireito would be effective in reducing frequent atrial tachyarrhythmias in the early-phase post-ablation.

### Methods

One hundred patients undergoing catheter ablation for symptomatic AF were randomly assigned to either a 30-day use of Saireito or control group. The primary endpoint was total number of episodes of frequent atrial tachyarrhythmias including definite recurrent AF and frequent premature atrial contractions during the 30-day treatment period post-ablation.

### Results

Three (6.0%) out of 50 patients treated with Saireito discontinued the drug because of adverse symptoms. The Saireito group was associated with a numerically lower number of episodes of frequent atrial tachyarrhythmias than the control group (3.1 versus 5.2 times, P = 0.17). The mean daily episodes of frequent atrial tachyarrhythmias were significantly fewer in the Saireito group during Day-6 to Day-10 (0.12/day versus 0.27/day, P = 0.03), and during Day-11 to Day-15 (0.08/day versus 0.24/day, P = 0.001). The prevalence of adverse symptoms during the 30-day treatment period was significantly higher in the Saireito group (18.0% versus 2.0%, P = 0.005).

**Funding:** The author(s) received no specific funding for this work.

**Competing interests:** The authors have declared that no competing interests exist.

**Abbreviations:** AAD, antiarrhythmic drug; AF, atrial fibrillation; ECG, electrocardiogram; PAC, premature atrial contraction.

## Conclusions

Thirty-day use of Saireito following AF ablation was associated with a tendency toward reduced number of episodes of frequent atrial tachyarrhythmias during the treatment period, with more pronounced effect in the first two weeks.

## Introduction

Catheter ablation has become popular as a curative therapy for atrial fibrillation (AF), but a sizable portion of patients have recurrent atrial tachyarrhythmias in the early phase following the procedure. The main cause of the early recurrence is considered to be the irritability and inflammation of the left atrium (LA) post-ablation. Previous studies have reported that approximately 30% of patients with early recurrence have no further arrhythmia recurrences beyond the unstable periods of the first couple of months post-ablation [1–3]. However, the early recurrence of AF can impair the quality-of-life of patients who received catheter ablation for AF [4]. Several studies have shown that short-term use of antiarrhythmic drugs (AADs) could reduce early arrhythmia recurrences, but the side effects of AADs are matters of concern [5–8].

The Saireito (TJ-114) is a Japanese herbal medicine, which is a combination of two herbs: Shosaikoto to calm inflammatory reactions and Goreisan to improve the water retention and edema. It has been reported that the Saireito improved acute or chronic inflammatory diseases such as rheumatoid arthritis, cirrhosis, nephrotic syndrome, and intestinal mucositis [9–11]. From these findings, we hypothesized that the Saireito was effective in reducing frequent atrial tachyarrhythmias in the early phase post ablation by calming the inflammation in the LA and improving the water retention. Accordingly, we conducted a dual-center prospective randomized controlled pilot trial, evaluating the efficacy and safety of the 30-day use of Saireito in patients undergoing catheter ablation for symptomatic AF.

## Methods

### Study design and protocol

The SAIreito Reduces Early recurrence after Intervention for aTrial fibrillation, Or not (SAIR-EITO) trial is a physician-initiated, non-company-sponsored, dual-center prospective randomized controlled pilot trial, comparing 30-day use of Saireito (N = 50) and a control (N = 50) in patients undergoing catheter ablation for symptomatic AF (*Japan Registry of Clinical Trials*: *jRCTs051200037*). The study protocol was approved by the institutional review board of Kyoto University Hospital and a participating institution (Mitsubishi Kyoto Hospital). Written informed consent was obtained from all the study patients. Patients with symptomatic AF undergoing first-time catheter ablation were recruited. Patients were eligible for the study if they were 20 to 85 years old, Exclusion criteria were contraindications or intolerance to Saireito including previous interstitial pneumonia, primary aldosteronism, an allergy to Saireito, and renal insufficiency (estimated glomerular filtration rate < 30 ml/min/1.73m$^2$ or on hemodialysis). Patients who had been treated with Saireito, Goreisan, and immunosuppressive drugs including steroids, those with a history of MAZE surgery or LA ablation, severe hepatic insufficiency, inability to be followed for one year, and unwillingness to sign the consent form for participation, and those whom the attending physician considered inappropriate to enroll in the study were also excluded. Between 5$^{th}$ October 2020 and 25$^{th}$ January 2022, 100 patients were enrolled in this trial.

The participants were randomly assigned to take Saireito or not in a 1:1 ratio before the ablation procedure. Randomization was performed via a computer-generated sequence using a permuted block design stratified by age ($<$ 70 years or $\geq$ 70 years), gender, center, AF type, AF duration ($<$ 3 years or $\geq$ 3 years), and type of the ablation procedure for pulmonary vein (PV) isolation (radiofrequency catheter ablation or cryoballoon ablation). In patients assigned to the 30-day use of Saireito, the drug was started on the day of the ablation procedure at a dose of 3 g for three times daily between meals. The operators were masked to the treatment assignment at the time of the ablation procedure, while the participants were not masked to the treatment assignment. For safety reasons, the participants were permitted to discontinue the intake of Saireito during the treatment period when side effects of Saireito were suspected.

## Ablation procedure

The ablation procedure was performed under the guidance of a three-dimensional mapping system (CARTO, Biosense-Webster, Diamond Bar, CA, USA; Ensite NavX, Abbott, Chicago, IL, USA; Rhythmia, Boston Scientific, Natick, MA, USA). PV isolation with radiofrequency catheter ablation was an extensive encircling PV isolation with two circular-shaped catheters placed in the ipsilateral superior and inferior pulmonary veins. After successful PV isolation, we checked whether dormant conduction between LA and PVs was provoked by adenosine triphosphate under isoproterenol infusion. When dormant conduction was induced, additional radiofrequency energy applications were delivered until disappearance of dormant conduction. In PV isolation with cryoballoon, single-shot freezing was performed for each PV, and electro-anatomical mapping with a high-density mapping catheter was performed to confirm the completion of PV isolation. When there was a conduction gap between PV and LA, touch-up ablation was performed using radiofrequency ablation catheter. Whether to perform additional ablation, such as tricuspid valve isthmus ablation, left atrial roof line ablation, mitral isthmus ablation, and superior vena cava isolation, was left to the discretion of the operators.

All AADs were discontinued before the ablation procedure, and restarted only when recurrent atrial tachyarrhythmias were detected. A second catheter ablation procedure was usually recommended to patients with recurrent atrial tachyarrhythmias after the blanking period of 3 months.

## Follow-up

Patients were scheduled to receive a periodical follow-up at the out-patient clinic of the centers where the index ablation was performed at 1-, 3-, 6-, and 12-month post ablation. A 12-lead electrocardiogram (ECG) was obtained at every visit. A one-channel ECG was recorded twice daily and at any time when the patient had symptoms with the use of an ambulatory electrogram recorder (HCG-801, OMRON HEALTHCARE Co., Ltd) for a duration of 1 month upon hospital discharge. Twenty-four-hour Holter-monitoring was performed at 3–6 months. The monitored ECGs during the index hospital admission were analyzed by the attending physicians. Ambulatory ECGs were read by cardiologists at the core laboratory. Holter-monitoring was read by clinicians at the local center. When patients became unable to visit the out-patient clinic of the local center, the follow-up data were obtained by contacting the physicians in charge or the patients. All data were evaluated and inputted by cardiologists or by the clinical research coordinators who were unaware of the treatment assignments.

## Definitions and outcome measures

AF was defined as paroxysmal when it terminated spontaneously or under AADs within 7 days of onset, and was considered persistent when it lasted for more than 7 days. The

European Heart Rhythm Association (EHRA) score was used to assess AF related symptoms: Grade 1 = none (excluded from the current study); Grade 2 = mild/moderate (normal daily activity not affected); Grade 3 = severe (normal daily activity affected); and Grade 4 = disabling (normal daily activity discontinued) [12].

The primary outcome measure was the total number of episodes of frequent atrial tachyarrhythmias defined as a composite of definite recurrent atrial tachyarrhythmias, frequent premature atrial contractions (PACs) (>5 per 30 beats), and short runs (>5 beats) of PACs, detected by the monitor or ambulatory ECG during the 30-day treatment period post ablation. The recurrent atrial tachyarrhythmias were defined as AF or atrial tachycardia (AT) lasting for >30 seconds or requiring repeat ablation, hospital admission, or usage of AADs. The secondary outcome measures were recurrent atrial tachyarrhythmias during the blanking period of 90-day, recurrent atrial tachyarrhythmias within 1-year after the blanking period of 90-day post-ablation, and Saireito-related adverse symptoms during the treatment period of 30 days. The Saireito-related adverse symptoms included a cough and digestive symptoms such as nausea, anorexia, and diarrhea. The serum cardiac enzymes such as the creatine kinase (CK), CK-MB and troponin I were measured before, and 6-hour, 1-day, 2-day, and 1-month after the procedure. The serum inflammatory biomarkers such as the white blood cells and C-reacting protein were measured before, and 1-day, 2-day, and 1-month after the procedure. The serum level of brain natriuretic peptides (BNP) and N-terminal pro BNP (NT-proBNP) were also measured before and 1-month after the procedure. Moreover, the extent of the body temperature elevation, body weight gain, and intravenous furosemide use during the index hospitalization were recorded. The quality of life was assessed by the Atrial Fibrillation Effect on QualiTy-of-life (AFEQT) score [13] before and 1-month, 3-month, and 1-year after the procedure.

## Statistical analysis

This was the first pilot study to evaluate the efficacy of Saireito in reducing early atrial tachyarrhythmias after AF ablation. Therefore, 50 patients in each group were set as the sample-size without calculation of the statistical power. The primary outcome measure, i.e., the total number of episodes of frequent atrial tachyarrhythmias during the treatment period of 1-month post ablation was presented as the mean value ± SD and compared using the Student $t$ test. Other data were presented as values and percentages, and the mean value ± SD or median with the first quartile to third quartile. Categorical variables were compared with the $\chi^2$ test or Fisher exact test. Continuous variables were compared using the Student $t$ test or Wilcoxon rank sum test based on their distribution. The cumulative incidence and event-free survival was estimated by the Kaplan-Meier method and the differences were assessed by the log-rank test. All analyses regarding the primary and secondary endpoints were performed by the intention-to-treat manner. Statistical analyses were performed using JMP 14 pro (SAS Institute Inc, Cary, NC) software. All the analyses were two-tailed, and a P value of <0.05 was considered statistically significant.

## Results

### Study patients

A total of 100 patients were enrolled between August 2020 and January 2022 in the study. Among the 50 patients assigned to the Saireito group, 3 (6.0%) prematurely discontinued the drug because of adverse symptoms (Fig 1). All study patients completed the 1-year follow-up, except for 1 patient in the control group who died due to colon cancer within a year.

# Study flow chart

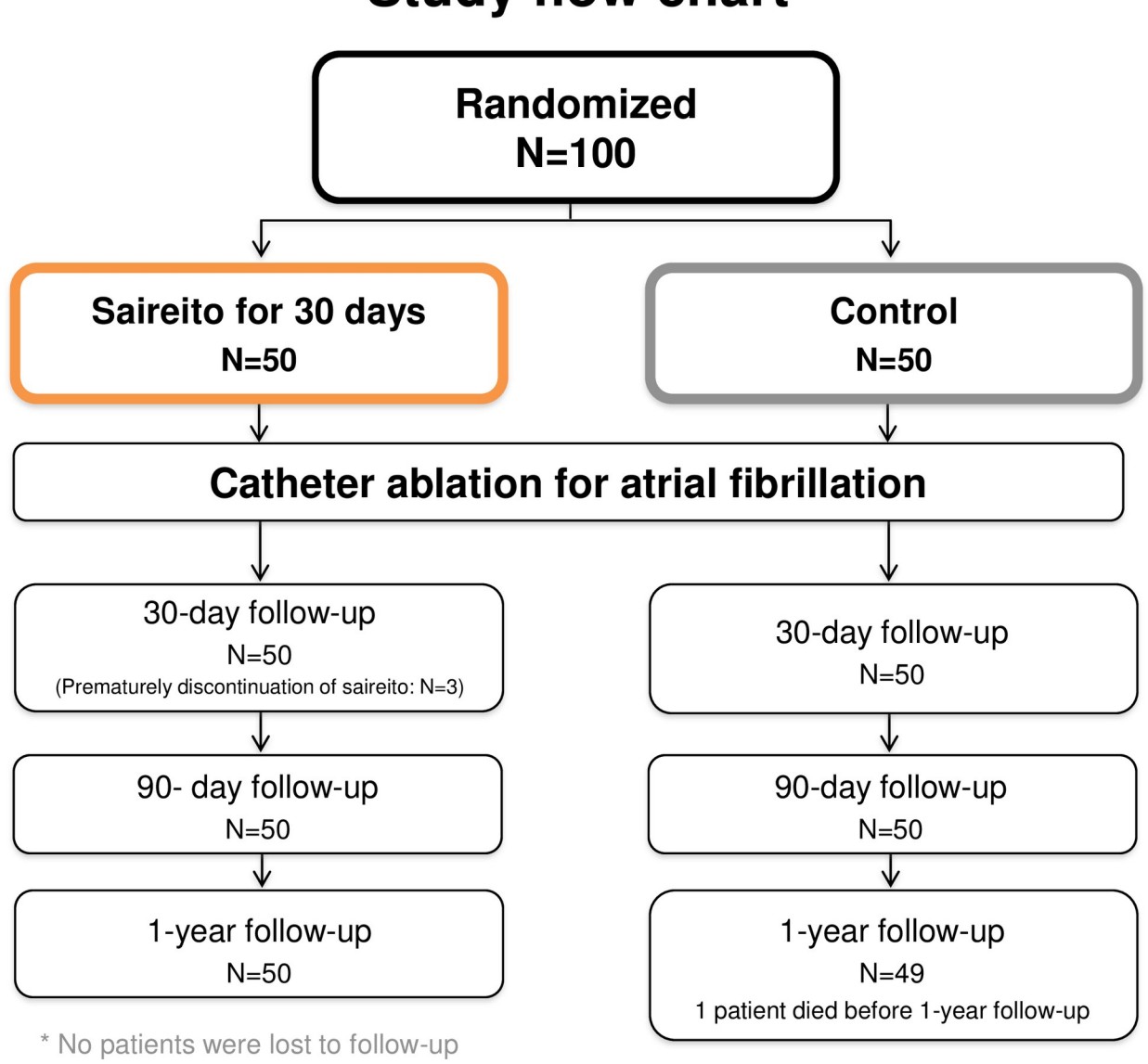

**Fig 1. Study flow chart.**

The baseline characteristics of the study patients are shown in Table 1. The average age was about 70 years and 82.0% of patients had paroxysmal AF. PV isolation was performed using cryoballoon in two-thirds of patients. All baseline characteristics were well balanced between the 2 groups.

## Clinical outcomes as an efficacy of Saireito-use

In the intention-to-treat analysis, the Saireito group was associated with a numerically smaller number of frequent atrial tachyarrhythmias as compared to the control group during the treatment period of 30 days (3.1±4.6 times versus 5.2±9.4 times, P = 0.17) (Fig 2A). Daily numbers of frequent atrial tachyarrhythmias during the 30 days after the procedure in the 2 groups are shown in Fig 3. Mean daily number of frequent atrial tachyarrhythmias during Day-6 to Day-

**Table 1. Baseline characteristics.**

| | Saireito N = 50 | Control N = 50 | P-value |
|---|---|---|---|
| Age (years old) | 70.6±10.0 | 69.7±9.6 | 0.68 |
| ≥70 | 29 (58.0%) | 31 (62.0%) | 0.68 |
| Female | 25 (50.0%) | 22 (44.0%) | 0.55 |
| Body weight (kg) | 61.4±10.5 | 62.5±12.8 | 0.63 |
| <60 | 26 (52.0%) | 26 (52.0%) | 1.00 |
| Paroxysmal atrial fibrillation | 41 (82.0%) | 41 (82.0%) | 0.34 |
| AF interval (years) | 0.7 (0.3–3.6) | 0.7 (0.3–3.2) | 0.55 |
| ≥3 | 14 (28.0%) | 14 (28.0%) | 1.00 |
| Symptom score (EHRA grade) | 3.0±0.8 | 3.0±0.7 | 0.90 |
| ≥3 | 34 (68.0%) | 36 (72.0%) | 1.00 |
| Hypertension | 32 (64.0%) | 25 (50.0%) | 0.16 |
| Diabetes | 3 (6.0%) | 3 (6.0%) | 1.00 |
| Previous heart failure | 3 (6.0%) | 2 (4.0%) | 0.65 |
| Chronic kidney disease (eGFR <60ml/min/1.73m$^2$) | 25 (50.0%) | 24 (48.0%) | 0.84 |
| CHADS$_2$ score | 1.3±1.2 | 1.2±1.0 | 0.41 |
| BNP (pg/ml) | 68.7 (34.7–154.5) | 73.5 (29.4–160.8) | 0.80 |
| ≥120 | 16 (32.7%) | 14 (31.8%) | 0.93 |
| NT-pro BNP (pg/ml) | 223.5 (108.5–614.8) | 210.5 (116.5–688.0) | 0.79 |
| ≥600 | 12 (26.1%) | 14 (29.2%) | 0.74 |
| Troponin I (pg/ml) | 2.7 (1.5–4.7) | 3.5 (1.7–7.7) | 0.23 |
| ≥5.0 | 10 (20.0%) | 17 (34.7%) | 0.10 |
| C-reacting protein (mg/dl) | 0.06 (0.03–0.11) | 0.07 (0.04–0.11) | 0.40 |
| ≥1.0 | 19 (38.0%) | 14 (28.0%) | 0.29 |
| Cryoballoon ablation | 34 (68.0%) | 32 (64.0%) | 0.67 |

Categorical variables are presented as numbers (percentage). Continuous variables are presented as the mean ± SD or median.

AF = atrial fibrillation; BNP = brain natriuretic peptides; eGFR = estimated glomerular filtration rate; NT-proBNP = N-terminal pro-brain natriuretic peptides

10 and Day-11 to Day-15 post-procedure was significantly lower in the Saireito group than in the control group (0.12 times/day and 0.27 times/day, P = 0.03, 0.08 times/day and 0.24 times/day, P = 0.001, respectively). Meanwhile, the cumulative incidence of the first episode of frequent atrial tachyarrhythmias did not differ between the 2 groups (68.0% versus 64.0%, P = 0.99) (Fig 2B). Furthermore, there was no difference between the groups in the event-free rates from definite recurrent atrial tachyarrhythmia within 90 days and after the blanking period of 90 days following the procedure (S1 Fig). The numerically lower number of the atrial tachyarrhythmias in the Saireito group during the treatment period was consistent in the patient subgroups (S1 Table).

The serum cardiac enzymes, such as CK, CK-MB, and Troponin I, peaked out at 6-hour after the procedure, and the levels did not significantly differ between the Saireito and control groups (Fig 4A). The inflammatory biomarkers, i.e., the serum white blood cells and C-reacting protein peaked out at Day-1 and Day-2, respectively, and the levels were comparable between the 2 groups (Fig 4B). The NT-proBNP level in the control group numerically increased immediately after the procedure but significantly decreased at 30-day, while that in

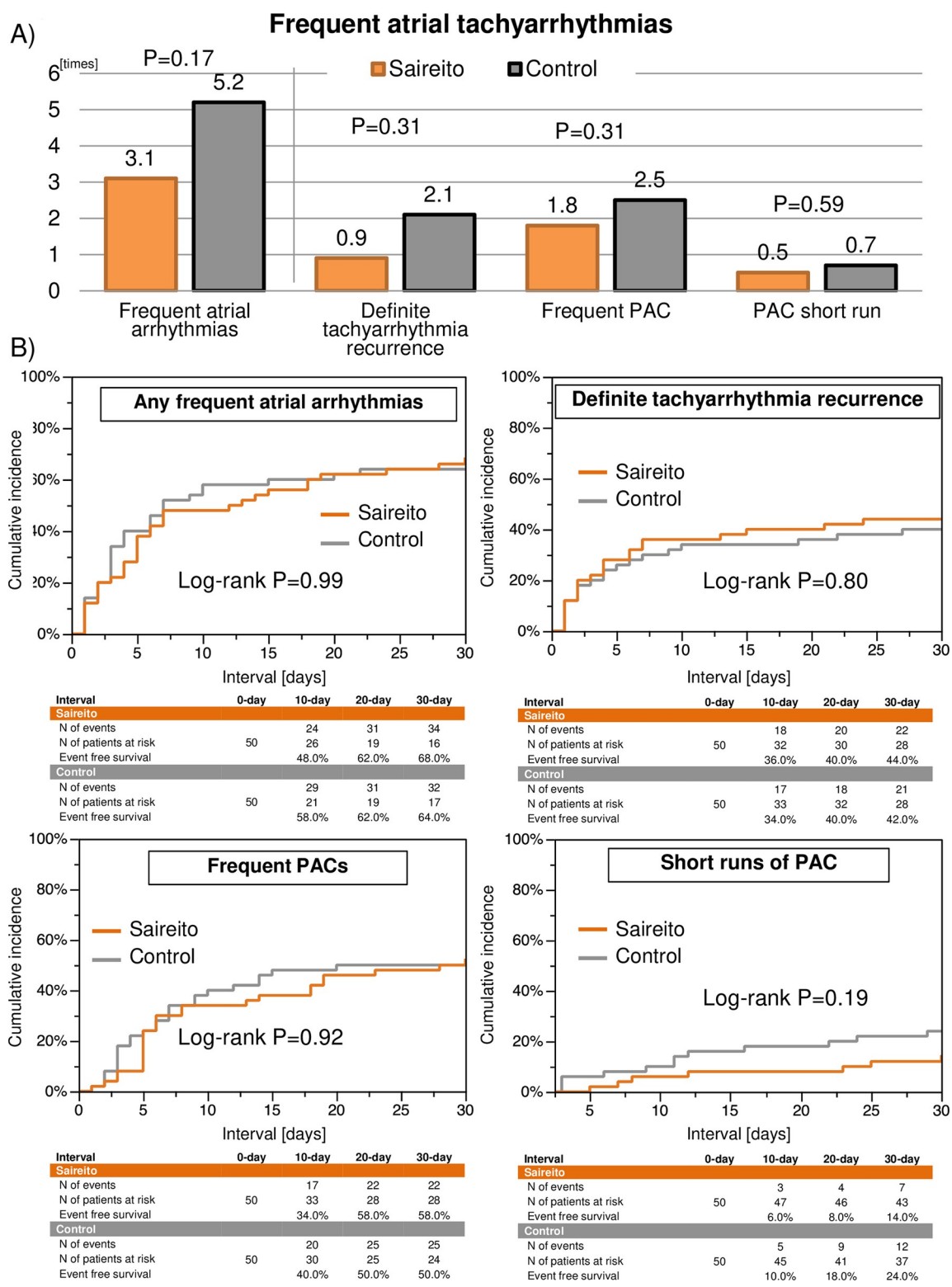

**Fig 2. Clinical outcome measures.** (A) Total episodes of frequent atrial tachyarrhythmias: a composite of definite recurrent atrial tachyarrhythmias, frequent premature atrial contractions (PACs), and PAC short runs during the treatment period of 30-day. (B) Cumulative incidence of the first episode of clinical outcomes.

# Timing of frequent atrial tachyarrhythmias

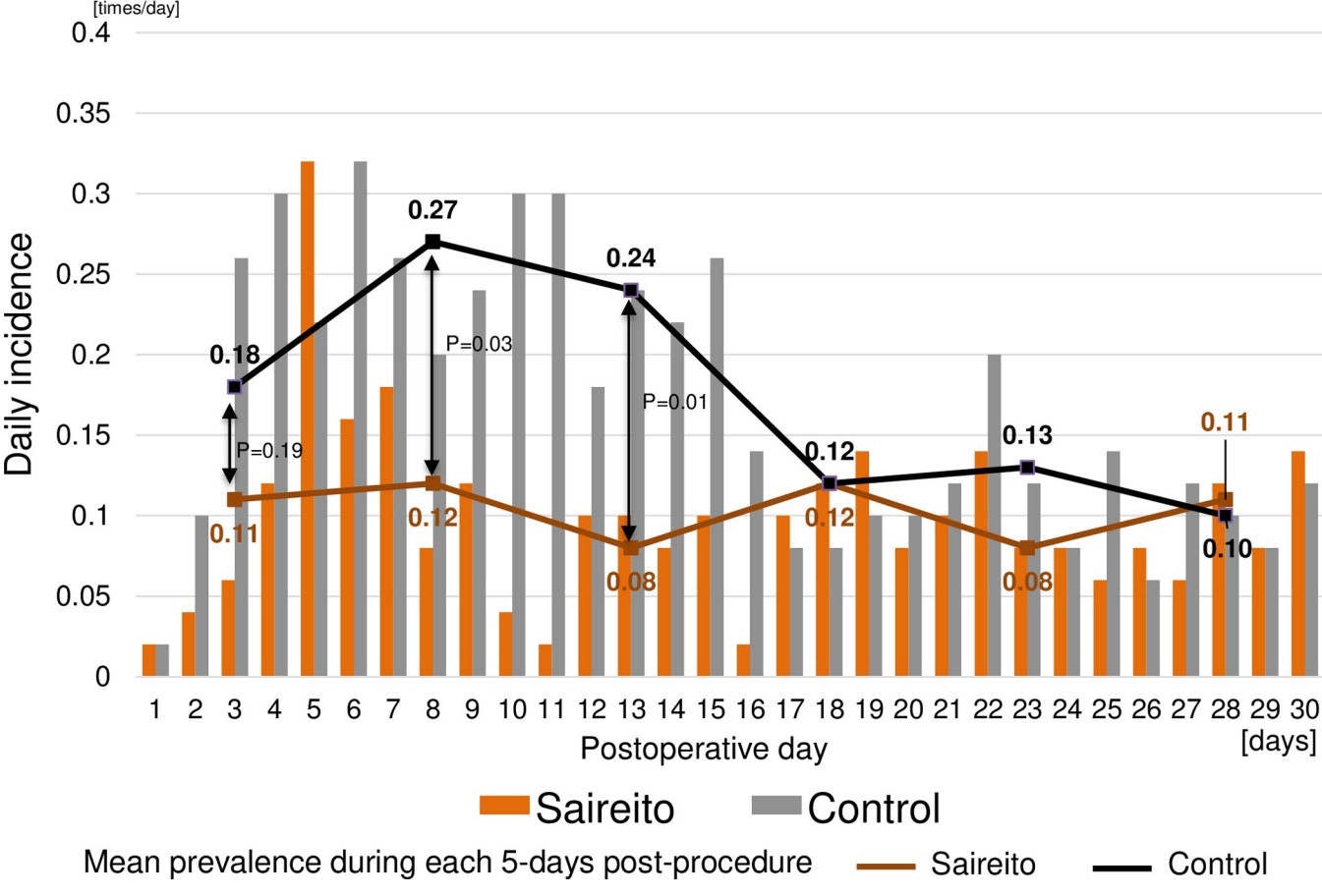

**Fig 3. Timing of frequent atrial tachyarrhythmias during the treatment period of 30 days.**

the Saireito group was stable during the 30-day treatment period (Fig 4C). The BNP level was comparable between the 2 groups during the treatment period. AFEQT score at 1-month post-procedure was significantly improved from baseline in both group (61.3 [51.7–76.0] versus 81.7 [69.6–89.0], P<0.001 in Saireito group; 62.1 [49.8–75.2] versus 80.4 [71.9–90.6], P<0.001 in control group), with no significant difference between the 2 groups (Fig 4D). The elevation in the body temperature during the index admission was significantly larger in the Saireito group than in the control group (Fig 4E). There was no significant difference in the extent of body weight gain and intravenous furosemide use between the 2 groups.

### Adverse outcomes as a measure of the safety of Saireito-use

The prevalence of adverse symptoms including a cough and digestive symptoms during the treatment period of 30 days was significantly higher in the Saireito group than in the control groups (18.0% versus 2.0%, P = 0.005) (Table 2). Three patients (6.0%) prematurely discontinued Saireito: 78-year-old patient at Day-12 for a fever and a cough, a 67-year-old female at Day-13 for nausea, and an 84-year-old male at Day-17 for a cough. The increase in the adverse symptoms during the treatment period was more pronounced in the elderly patients (≥70 years old), males, or early (<3 years) AF patients with no heart failure, chronic kidney disease, or with high BNP levels (S1 Table).

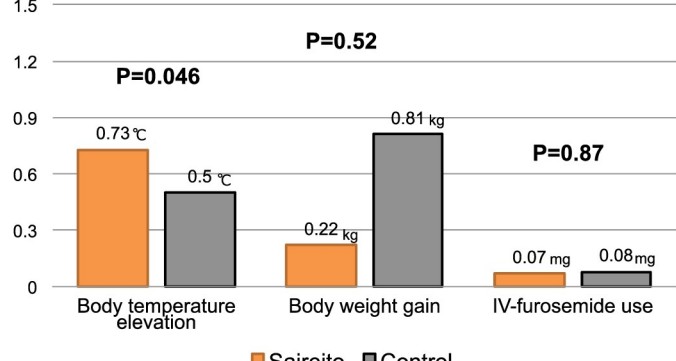

**Fig 4. Serum biomarkers and quality of life after the procedure.** A) Cardiac enzymes, B) Inflammatory biomarkers, C) Natriuretic peptides, D) QOL score, and E) clinical outcomes during the index admission. BNP = brain natriuretic peptides; CK = creatine kinase; CRP = C-reacting protein; IV = intravenous; NT-proBNP = N-terminal pro-brain natriuretic peptides; QOL = quality of life; TnI = troponin I; WBC = white blood cell.

Interstitial pneumonia occurred in 2 patients (4.0%) who had had no obvious lung disease and had completed the 30-day use of Saireito: an 83-year-old male with an onset at Day-31 and a 74-year-old male with an onset at Day-40 post procedure (S2 Fig). Both patients received

**Table 2. Adverse clinical symptoms over 30 days.**

| | Saireito<br>N = 50 | Control<br>N = 50 | P-value |
|---|---|---|---|
| Any adverse symptoms | 9 (18.0%) | 1 (2.0%) | 0.008 |
| Cough | 4 (8.0%) | 0 (0.0%) | 0.06 |
| Digestive symptoms<br>(nausea, anorexia, diarrhea) | 5 (10.0%) | 1 (2.0%) | 0.10 |

Categorical variables are presented as numbers (percentage).

short-term steroid therapy and fully recovered from the pneumonia. No pseudoaldosteronism or new-onset severe liver failure as a side effect of Saireito did not occur during the study period.

## Discussion

To the best of our knowledge, the present randomized pilot trial was the first to evaluate the efficacy and safety of a 30-day use of Saireito after catheter ablation for symptomatic AF. The main findings were as follows: 1) 30-day use of Saireito following catheter ablation for AF numerically, but not significantly, reduce the frequent atrial tachyarrhythmias during the treatment period as compared to the control group; 2) Mean daily number of episodes of the frequent atrial tachyarrhythmias during the first two weeks in the Saireito group was significantly lower than in the control group; 3) an acute inflammatory reaction post-procedure was comparable between the Saireito and control groups; and 4) 30-day use of Saireito was associated with a higher prevalence of adverse symptoms during the treatment period.

Early arrhythmia recurrence is often associated with late recurrences beyond 3-month blanking period post AF ablation. Therefore, several randomized clinical trials have evaluated the efficacy of short-term use of AADs after AF ablation in suppressing early and late arrhythmia recurrences. The Antiarrhythmics After Ablation of Atrial Fibrillation (5A Study) trial reported that 6-week use of AADs significantly reduced the arrhythmia recurrences during the treatment period [5, 6]. Also in the AMIOdarone after CATheter ablation for atrial fibrillation (AMIO-CAT) trials, 8-week use of amiodarone significantly reduced early recurrences as compared to the control group [7]. Furthermore, a more recent and larger trial, the Efficacy of Anti arrhythmic drugs Short-Term use after catheter ablation for Atrial Fibrillation (EAST-AF) trial enrolled 2038 AF patients, showing that 90-day use of AADs significantly reduced early recurrences [8]. However, all those trials reported that successful suppression of early arrhythmia recurrences by AADs did not lead to lower rate of late arrhythmia recurrences beyond the treatment period. We for the first time assessed the efficacy of Saireito in reducing early atrial tachyarrhythmias post ablation, but could not demonstrate the significant utility of the drug.

Saireito is a combination of Shosaikoto and Goreisan, and is expected to calm inflammatory reactions and improve the water retention. Kaneko T, et al. reported that Saireito inhibited the inducible nitro oxide expression and prostaglandin $E_2$ production in lipopolysaccharide-stimulated cells, leading to anti-inflammatory activity [14]. Ono T, et al. found that Saireito suppressed the major profibrotic factors transforming growth factor (TGF)-βand connective tissue growth factor (CTGF) in rat mesangioproliferative glomerulonephritis [15]. Moreover, Kato S, et al. demonstrated that Saireito attenuated the up-regulation of TNF-α and IL-1βmRNA in 5-FU-induced intestinal mucositis of mice [11]. Furthermore, several small studies reported that Saireito reduced the acute edema and inflammatory reactions after total hip

arthroplasty, radiotherapy, and acquired ptosis surgery [16–18]. Atrial tachyarrhythmias early after catheter ablation for AF are considered to be caused by the irritability of LA from inflammation due to the ablation and water retention. Therefore, we hypothesized that short-term use of Saireito was effective in reducing early atrial tachyarrhythmias after AF ablation. However, probably due to the small number of patients enrolled in this pilot study, Saireito numerically reduced atrial tachyarrhythmias post ablation, but the difference was not statistically significant. The efficacy of short-term usage of Saireito after AF ablation should be evaluated in future larger randomized clinical trials.

Regarding the safety of Saireito-use, interstitial pneumonia, which is one of the side effects of Ogon (a component of Saireito), occurred in 2 elderly patients (4.0%) in the Saireito group. Given the very low prevalence of spontaneous interstitial pneumonia (<0.1%) and the timing of the onset of the pneumonia in those 2 cases (within 10 days after the treatment period), causal relationships between Saireito use and interstitial pneumonia were strongly suspected. The physicians should be cautious of interstitial pneumonia with Saireito use especially in elderly patients. Considering the pronounced effect of Saireito in reducing atrial tachyarrhythmias in the first 2 weeks after ablation and the risk of the side effects, especially interstitial pneumonitis, 2 weeks rather than 30 days may be better for the treatment period of Saireito. The prevalence of adverse symptoms, i.e., coughs and digestive symptoms were more frequent in the Saireito group (18.0%) than in the control group (2.0%), leading to discontinuation of the drug in 3 patients (6.0%) in the present study. However, in the acute phase after AF ablation, such symptoms are commonly observed and may be a sign of iatrogenic pulmonary stenosis, pericardial effusion, or acute gastric hypomotility [19–21]. Therefore, newly added medications after AF ablation are likely to be discontinued for those adverse symptoms.

## Limitations

There were several important limitations in this study. First, the current study did not have a statistical power for the primary outcome because of the small sample size, which precluded any definitive conclusions. However, we for the first time evaluated the efficacy and safety of the 30-day use of Saireito after catheter ablation for symptomatic AF in this pilot trial. Second, the assigned treatment was blinded only to the operators during the ablation procedure, but not to the participants, because of unavailability of the placebo of Saireito. Patients' wariness toward newly started Saireito after ablation might have affected the adverse symptoms. Third, implantable loop recorder was not used in this study. Although the patients received intensive monitoring with an ambulatory electrogram recorder during 30-day treatment period after the procedure, there may have been substantial undetected episodes of frequent atrial tachyarrhythmias during follow-up.

## Conclusions

Thirty-day use of Saireito following catheter ablation for AF showed a trend toward reduced atrial tachyarrhythmias during the treatment period. The favorable effect was more pronounced and significant in the first 2 weeks post ablation. Despite short-term use, interstitial pneumonia occurred in 2 patients (4.0%) in the Saireito group early after the treatment period. Thus, two-week use of Saireito seems reasonable, but its efficacy and safety should be evaluated in future larger randomized clinical trials.

## Supporting information

**S1 Table. Clinical outcomes in the subgroups.**
(DOCX)

**S1 Fig. Event-free survival from definite recurrent atrial tachyarrhythmias within and after 90-days.**
(TIF)

**S2 Fig. Two cases of interstitial pneumonia in the Saireito group.**
(TIF)

## Author Contributions

**Data curation:** Tetsuma Kawaji, Fumiya Yoneda, Shushi Nishiwaki, Munekazu Tanaka, Takanori Aizawa, Shun Hojo, Kenji Nakatsuma.

**Investigation:** Tetsuma Kawaji.

**Methodology:** Tetsuma Kawaji, Hidenori Yaku, Kazuhisa Kaneda.

**Supervision:** Satoshi Shizuta, Masashi Kato, Takafumi Yokomatsu, Shinji Miki, Koh Ono, Takeshi Kimura.

**Writing – original draft:** Tetsuma Kawaji.

**Writing – review & editing:** Satoshi Shizuta.

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
