## [Decision Letter · Decision Letter 0]

29 May 2024

PONE-D-23-41672Efficacy and safety of Saireito (TJ-114) in patients with atrial fibrillation undergoing catheter ablation procedures: a randomized pilot studyPLOS ONE

Dear Dr. Shizuta,

Thank you for submitting your manuscript to PLOS ONE. After careful consideration, we feel that it has merit but does not fully meet PLOS ONE’s publication criteria as it currently stands. Therefore, we invite you to submit a revised version of the manuscript that addresses the points raised during the review process.

We look forward to receiving your revised manuscript.

Kind regards,

Ibrahim Marai, MD

Academic Editor

PLOS ONE

[None]. 

3. Please review your reference list to ensure that it is complete and correct. If you have cited papers that have been retracted, please include the rationale for doing so in the manuscript text, or remove these references and replace them with relevant current references. Any changes to the reference list should be mentioned in the rebuttal letter that accompanies your revised manuscript. If you need to cite a retracted article, indicate the article’s retracted status in the References list and also include a citation and full reference for the retraction notice

Reviewers' comments:

Reviewer's Responses to Questions

**Comments to the Author**

1. Is the manuscript technically sound, and do the data support the conclusions?

Reviewer #1: Yes

2. Has the statistical analysis been performed appropriately and rigorously? 

Reviewer #1: Yes

3. Have the authors made all data underlying the findings in their manuscript fully available?

Reviewer #1: Yes

4. Is the manuscript presented in an intelligible fashion and written in standard English?

Reviewer #1: Yes

5. Review Comments to the Author

Reviewer #1: I have read with interest the manuscript titled " Efficacy and safety of Saireito (TJ-114) in patients with atrial fibrillation undergoing catheter ablation procedures: a randomized pilot study”. The authors conducted a randomized placebo-controlled prospective single-blinded clinical trial analyzing the anti-inflammatory and anti-edema effects of the Japanese herbal medicine Saireito post-catheter ablation of atrial fibrillation. In particular, in this pilot study they analyzed whether the intake of this traditional herbal medicine reduced the burden of frequent atrial arrhythmias in the 30 days after the procedure (primary endpoint). This period is within the so-called blanking period, in which the burden of atrial arrhythmias might paradoxically increase due to the fact that ablation lesions have not yet completely stabilized and usually is not taken into account in clinical trials, however patients still might experience discomfort due to symptoms with quality-of-life reduction and might necessitate ER visits and/or hospitalization. All anti-arrhythmic drugs were discontinued before ablation procedure in order to avoid bias and analyze Saireito true effect. The authors demonstrated that at the end of the follow up Saireito did not reduce atrial arrhythmias in a statistically significant fashion, however there was a tendency in this reduction, more pronounced in the first two weeks, and trials with bigger sample sizes might provide better insights on the topic. Moreover the use of Saireito was associated with adverse events, in particular with a non-negligible rate of interstitial pneumonia (4%, i.e. two patients).

I think this manuscript is well written and does not have any major issue in form or design of the study. The study design and the endpoints are clear, as well as inclusion and exclusion criteria. The topic is debated among experts and the use of this Herbal Medicine might be a novelty. I therefore wholeheartedly congratulate the Authors for their work.

I would like to highlight the following aspects:

1. The study is single-blinded for the operators, participants were not masked to the treatment assignment. The reason of this choice should be clearly stated.

2. Ablation procedures, both using RF and cryoballoon, have a plethora a possible variables to be considered to define ablation success. However, maybe due to word limit, it was not clearly defined how ablation success was evaluated. Please elaborate on this and provide the exact variables used to define clinical success. Furthermore were mapping systems used? Was remapping after energy delivery evaluated?

3. In the “Methods – Follow up” section I would suggest to change the sentence “The monitored ECGs during the index admission were read by the attending physicians”, In particular I would suggest to use the word “analyze” in lieu of read, to better convey the message I think the Author intended to pass on.

4. A 30-day period was chosen. Why did the Author chose such period? I think it would be interesting to analyze the effect of Saireito on the whole blanking period, which is conventionally defined as 90 days for most trials, or at least 45 days, as some authors advocates for shorter blanking periods.

5. Was the use of Implantable Loop Recorders (ILRs) to detect atrial arrhythmias taken into consideration? I think the use of ILRs could have provided invaluable data on this matter, virtually eliminating the possibility of undetected episodes of atrial arrhythmias (a limit clearly stated by the Authors on the manuscript)

6. I think the abstract conclusion section should be changed to avoid misleading the readers, in particular I would change the part “but the favorable effect was remarkable in the first 2 weeks” with “there was a tendency in their reduction, more pronounced in the first two weeks”.

7. Last but not least, the use of Saireito was associated with adverse events, in particular with a non-negligible rate of interstitial pneumonia (4%, i.e. two patients) which could theoretically lead to further morbidity and mortality and have long lasting effects on the patients. Did the two patients had any long-lasting effects due to pneumonia? Furthemore, could you elaborate more on this aspect and in particular on its pathogenesis? Moreover, considering that, as you cited, interstitial pneumonia is a side effects of Ogon, a component of Saireito, is it possible and/or plausible to avoid its use in the Saireito compound? Considering this non-negligible rate of this complication, are further studies using Saireito ethically viable in your opinion?

I congratulate again the Authors for their invaluable work.

6. PLOS authors have the option to publish the peer review history of their article (what does this mean?). If published, this will include your full peer review and any attached files.

Reviewer #1: **Yes: **Domenico Pecora

---

## [Author Response · Author response to Decision Letter 0]

10 Jul 2024

Response to Reviewers

We deeply appreciate the editor and the reviewer for the critically important comments and suggestions on our paper. We have revised our manuscript according to those comments. 

All essential changes in the revised manuscript were highlighted in red font. 

Our replies to the comments and suggestions of the editor and the reviewer are written below.

Reviewer #1: I have read with interest the manuscript titled " Efficacy and safety of Saireito (TJ-114) in patients with atrial fibrillation undergoing catheter ablation procedures: a randomized pilot study”. The authors conducted a randomized placebo-controlled prospective single-blinded clinical trial analyzing the anti-inflammatory and anti-edema effects of the Japanese herbal medicine Saireito post-catheter ablation of atrial fibrillation. In particular, in this pilot study they analyzed whether the intake of this traditional herbal medicine reduced the burden of frequent atrial arrhythmias in the 30 days after the procedure (primary endpoint). This period is within the so-called blanking period, in which the burden of atrial arrhythmias might paradoxically increase due to the fact that ablation lesions have not yet completely stabilized and usually is not taken into account in clinical trials, however patients still might experience discomfort due to symptoms with quality-of-life reduction and might necessitate ER visits and/or hospitalization. All anti-arrhythmic drugs were discontinued before ablation procedure in order to avoid bias and analyze Saireito true effect. The authors demonstrated that at the end of the follow up Saireito did not reduce atrial arrhythmias in a statistically significant fashion, however there was a tendency in this reduction, more pronounced in the first two weeks, and trials with bigger sample sizes might provide better insights on the topic. Moreover the use of Saireito was associated with adverse events, in particular with a non-negligible rate of interstitial pneumonia (4%, i.e. two patients).

I think this manuscript is well written and does not have any major issue in form or design of the study. The study design and the endpoints are clear, as well as inclusion and exclusion criteria. The topic is debated among experts and the use of this Herbal Medicine might be a novelty. I therefore wholeheartedly congratulate the Authors for their work.

I would like to highlight the following aspects:

1. The study is single-blinded for the operators, participants were not masked to the treatment assignment. The reason of this choice should be clearly stated.

Thank you for your important comments. As you indicated, the assigned treatment was blinded only to the operators during the ablation procedure, but not to the patients. This is an important limitation of the study. We added the following sentences in the Limitation section.

<Limitations>

Second, the assigned treatment was blinded only to the operators during the ablation procedure, but not to the participants, because of unavailability of the placebo of Saireito. Patients’ wariness toward newly started Saireito after ablation might have affected the adverse symptoms. (Page 11, Line 4- Page 11, Line 7)

2. Ablation procedures, both using RF and cryoballoon, have a plethora a possible variables to be considered to define ablation success. However, maybe due to word limit, it was not clearly defined how ablation success was evaluated. Please elaborate on this and provide the exact variables used to define clinical success. Furthermore were mapping systems used? Was remapping after energy delivery evaluated?

Thank you for your comment. Our main strategy in AF ablation was pulmonary vein (PV) isolation. Whether to perform additional ablations were left to the discretion of the operators. When PV isolation was performed by radiofrequency catheter ablation, adenosine-triphosphate test was performed to check dormant conduction. When cryoballoon was used for PV isolation, 3D-electrioanatomical mapping was performed to confirm the completion of PV isolation. We added the following sentences in the Methods section.

<Methods>

The ablation procedure was performed under the guidance of a three-dimensional mapping system (CARTO, Biosense-Webster, Diamond Bar, CA, USA; Ensite NavX, Abbott, Chicago, IL, USA; Rhythmia, Boston Scientific, Natick, MA, USA). PV isolation with radiofrequency catheter ablation was an extensive encircling PV isolation with two circular-shaped catheters placed in the ipsilateral superior and inferior pulmonary veins. After successful PV isolation, we checked whether dormant conduction between LA and PVs was provoked by adenosine triphosphate under isoproterenol infusion. When dormant conduction was induced, additional radiofrequency energy applications were delivered until disappearance of dormant conduction. In PV isolation with cryoballoon, single-shot freezing was performed for each PV, and electro-anatomical mapping with a high-density mapping catheter was performed to confirm the completion of PV isolation. When there was a conduction gap between PV and LA, touch-up ablation was performed using radiofrequency ablation catheter. Whether to perform additional ablation, such as tricuspid valve isthmus ablation, left atrial roof line ablation, mitral isthmus ablation, and superior vena cava isolation, was left to the discretion of the operators. (Page 4, Line 21 – Page 21, Line 7)

3. In the “Methods – Follow up” section I would suggest to change the sentence “The monitored ECGs during the index admission were read by the attending physicians”, In particular I would suggest to use the word “analyze” in lieu of read, to better convey the message I think the Author intended to pass on.

Thank you very much for your kind comment. As you suggested, we revised the following sentences in the Methods section.

<Methods>

Twenty-four-hour Holter-monitoring was performed at 3-6 months. The monitored ECGs during the index hospital admission were analyzed by the attending physicians. (Page 5, Line 17 – Page 5, Line 18)

4. A 30-day period was chosen. Why did the Author chose such period? I think it would be interesting to analyze the effect of Saireito on the whole blanking period, which is conventionally defined as 90 days for most trials, or at least 45 days, as some authors advocates for shorter blanking periods.

Thank you for your important comment. We agree that the 90-day period is a gold standard for the blanking-period after catheter ablation for AF. Recently, however, a shorter blanking period of 2-month has been also proposed (Heart Rhythm 2024;21:521–529, J Cardiovasc Electrophysiol. 2020;31:2363–2370). Furthermore, in the recent report by Onishi et al (Int J Cardiol. 2021 Oct 15:341:39-45), patients with early arrhythmia recurrences between 1- and 3-month post ablation was associated with much higher rate of late arrhythmia recurrence beyond the blanking period of 90 days, as compared to those with early arrhythmia recurrences within 1 month. 

Also considering the risk of potential side effects of Saireito, we chose relatively short treatment period of 30 days in this pilot study.

5. Was the use of Implantable Loop Recorders (ILRs) to detect atrial arrhythmias taken into consideration? I think the use of ILRs could have provided invaluable data on this matter, virtually eliminating the possibility of undetected episodes of atrial arrhythmias (a limit clearly stated by the Authors on the manuscript)

We totally agree with your opinion. We wanted to use ILRs to precisely assess the burden of early atrial tachyarrhythmias after procedure in this study. However, in Japan, ILRs were officially covered by medical insurance only for syncope and embolic stroke of unknown sources (ESUS). Also, the present study was non-company-sponsored physician-initiated study. Therefore, we used the ambulatory ECG monitors instead of ILRs, which is a limitation of the study. We added descriptions regarding this point in the limitation section, as pasted below.

<Limitations>

Third, implantable loop recorder was not used in this study. Although the patients received intensive monitoring with an ambulatory electrogram recorder during 30-day treatment period after the procedure, there may have been substantial undetected episodes of frequent atrial tachyarrhythmias during follow-up. (Page 11, Line 7 – Page 11, Line 10)

6. I think the abstract conclusion section should be changed to avoid misleading the readers, in particular I would change the part “but the favorable effect was remarkable in the first 2 weeks” with “there was a tendency in their reduction, more pronounced in the first two weeks”.

Thank you for your kind comment. We revised the conclusion section of the abstract according to your suggestion, as pasted below.

<Abstract>

Conclusions Thirty-day use of Saireito following AF ablation was associated with a tendency toward reduced number of episodes of frequent atrial tachyarrhythmias during the treatment period, with more pronounced effect in the first two weeks. (Page 2, Line 16 - Page 2, Line 18)

7. Last but not least, the use of Saireito was associated with adverse events, in particular with a non-negligible rate of interstitial pneumonia (4%, i.e. two patients) which could theoretically lead to further morbidity and mortality and have long lasting effects on the patients. Did the two patients had any long-lasting effects due to pneumonia? Furthemore, could you elaborate more on this aspect and in particular on its pathogenesis? Moreover, considering that, as you cited, interstitial pneumonia is a side effects of Ogon, a component of Saireito, is it possible and/or plausible to avoid its use in the Saireito compound? Considering this non-negligible rate of this complication, are further studies using Saireito ethically viable in your opinion?

I congratulate again the Authors for their invaluable work.

Thank you for your comment. Interstitial pneumoniae (IP) is a main safety concern for the use of Saireito. It is considered a side effect of Ogon, a component of Saireito. Unfortunately, it is difficult to remove Ogon from Sareito. The two patients with IP received short-term steroid therapy and fully recovered from IP without any long-lasting effects. 

We added the following sentences in the Results and Discussion sections.

<Results>

Interstitial pneumonia occurred in 2 patients (4.0%) who had had no obvious lung disease and had completed the 30-day use of Saireito: an 83-year-old male with an onset at Day-31 and a 74-year-old male an onset at Day-40 post procedure (Figure S2). Both patients received short-term steroid therapy, and fully recovered from the pneumonia. No pseudoaldosteronism or new-onset severe liver failure as a side effect of Saireito did not occur during the study period. (Page 8, Line 24 – Page 8, Line 28)

<Discussion>

Regarding the safety of Saireito-use, interstitial pneumonia, which is one of the side effects of Ogon (a component of Saireito), occurred in 2 elderly patients (4.0%) in the Saireito group. Given the very low prevalence of spontaneous interstitial pneumonia (<0.1%) and the timing of the onset of the pneumonia in those 2 cases (within 10 days after the treatment period), causal relationships between Saireito use and interstitial pneumonia were strongly suspected. The physicians should be cautious of interstitial pneumonia with Saireito use especially in elderly patients. Considering the pronounced effect of Saireito in reducing atrial tachyarrhythmias in the first 2 weeks after ablation and the risk of side effects of Saireito, especially interstitial pneumonitis, 2 weeks rather than 30 days may be better to maximize the benefit and minimize the risk of Saireito use after AF ablation. The prevalence of adverse symptoms, i.e., coughs and digestive symptoms were more frequent in the Saireito group (18.0%) than in the control group (2.0%), leading to discontinuation of the drug in 3 patients (6.0%) in the present study. However, in the acute phase after AF ablation, such symptoms are commonly observed and may be a sign of iatrogenic pulmonary stenosis, pericardial effusion, or acute gastric hypomotility.19-21 Therefore, newly added medications after AF ablation are likely to be discontinued for those adverse symptoms. (Page 10, Line 14 – Page 10, Line 25)

---

## [Editor Report · Decision Letter 1]

12 Jul 2024

Efficacy and safety of Saireito (TJ-114) in patients with atrial fibrillation undergoing catheter ablation procedures: a randomized pilot study

PONE-D-23-41672R1

Dear Dr.Satoshi Shizuta

We’re pleased to inform you that your manuscript has been judged scientifically suitable for publication and will be formally accepted for publication once it meets all outstanding technical requirements.

Kind regards,

Ibrahim Marai, MD

Academic Editor

PLOS ONE

---

## [Editor Report · Acceptance letter]

23 Jul 2024

PONE-D-23-41672R1 

PLOS ONE

Dear Dr. Shizuta, 

I'm pleased to inform you that your manuscript has been deemed suitable for publication in PLOS ONE. Congratulations! Your manuscript is now being handed over to our production team.

Kind regards, 

on behalf of

Dr. Ibrahim Marai 

Academic Editor

PLOS ONE